# Preparation, Isolation and Antioxidant Function of Peptides from a New Resource of *Rumexpatientia L.* ×*Rumextianshanicus A. Los*

**DOI:** 10.3390/foods13070981

**Published:** 2024-03-22

**Authors:** Chang Liu, Jianing Wang, Dan Hong, Zhou Chen, Siting Li, Aijin Ma, Yingmin Jia

**Affiliations:** School of Food and Health, Beijing Technology and Business University, Beijing 100048, China; lc15373689836@163.com (C.L.); 17332206521@163.com (J.W.); 15933576863@163.com (D.H.); zhouch2017@btbu.edu.cn (Z.C.); lisiting@btbu.edu.cn (S.L.); maaj@btbu.edu.cn (A.M.)

**Keywords:** *Rumexpatientia L.* ×*Rumextianshanicus A. Los*, antioxidant peptides, purification, identification, H_2_O_2_, HepG2

## Abstract

*Rumexpatientia L.* ×*Rumextianshanicus A. Los* (RRL), known as “protein grass” in China, was recognized as a new food ingredient in 2021. However, the cultivation and product development of RRL are still at an early stage, and no peptide research has been reported. In this study, two novel antioxidant peptides, LKPPF and LPFRP, were purified and identified from RRL and applied to H_2_O_2_-induced HepG2 cells to investigate their antioxidant properties. It was shown that 121 peptides were identified by ultrafiltration, gel filtration chromatography, and LC-MS/MS, while computer simulation and molecular docking indicated that LKPPF and LPFRP may have strong antioxidant properties. Both peptides were not cytotoxic to HepG2 cells at low concentrations and promoted cell growth, which effectively reduced the production of intracellular ROS and MDA, and increased cell viability and the enzymatic activities of SOD, GSH-Px, and CAT. Therefore, LKPPF and LPFRP, two peptides, possess strong antioxidant activity, which provides a theoretical basis for their potential as food additives or functional food supplements, but still need to be further investigated through animal models as well as cellular pathways.

## 1. Introduction

*Rumexpatientia L.* ×*Rumextianshanicus A. Los* (RRL), which is called “protein grass” in China, belongs to the Polygonaceae family of plants in the genus *Rumex*. The genus *Rumex* consists of over 200 species, which are mainly found in the northern temperate zone. There are many important secondary metabolites, including anthraquinones, flavonoids, stilbenes, naphthalenes, and terpenes [1,2]. In folklore, the leaves, flowers, and seeds of certain *Rumex* plant species are consumed as vegetables. Recent studies have shown that pure substances and extracts from different *Rumex* plants have a wide range of biological activities, such as protecting the liver, fighting viruses and tumors, fighting cancer, boosting the immune system, reducing inflammation, and protecting against radiation. It may also be used to prevent a variety of diseases, including constipation, diarrhea, diabetes mellitus, hypertension, and cardiovascular disease [1,3,4,5,6].

RRL has received significant attention in China due to its high yield, long lifespan, adaptability, and wide distribution. It was obtained by hybridizing *Rumex K-1* (*Rumex patientia ×Rumex tianschaious*) with *Rumex crispus L.* in China during the 1970s [7]. RRL has a history of approximately twenty years of food consumption in China and has been used in various fields, including food, forage, and the environment, in the last decade [8]. RLL was recognized as a new food ingredient by the National Health Commission of the PRC through the provisions of *China’s Food Safety Law* in 2021, in accordance with the standards for *green leafy vegetables* (NY/T 743-2020). The protein content of RRL dry matter reaches 30~48.7% and is rich in 18 amino acids, including high levels of Glu, Asp, and Leu [9], while lower levels of Cys and Met are the limiting amino acids of RRL. It contains all essential amino acids, approximately 45 percent of total amino acids, which can adequately meet the Food and Agriculture Organization of the United Nations/World Health Organization’s (FAO/WHO) recommended intake of essential amino acids for adults [10]. It is a high-quality source of protein, and at the same time, it can be a biologically active peptide from a good source. The consumption of RRL can be accompanied by foods rich in Cys and Met to enhance the overall use of the nutritional value. In recent years, scholars worldwide have shown great interest in plant-derived active peptides as antioxidants. Antioxidant-rich peptides may have additional benefits such as anti-aging, improved memory activity, and anti-inflammatory effects [11]. Therefore, there is a wide market prospect for developing RRL to obtain novel, strong antioxidant peptides and apply them to functional foods. This has significant value in the food industry, particularly in the functional food sector.

Currently, the cultivation and product development of RRL are still at the primary stage, and no studies on its peptides have been reported. Although a study has reported that the extract of RRL leaves had antioxidant activity, the active components involved could not be identified [10]. Therefore, this study aimed to purify and identify two novel antioxidant peptides, LKPPF and LPFRP, from RRL. The study used H_2_O_2_-induced HepG2 cells to provide theoretical support for investigating the antioxidant effects of RRL peptides and to establish a foundation for further exploring their antioxidant mechanism.

## 2. Materials and Methods

### 2.1. Materials

*Rumexpatientia L. ×Rumextianshanicus A. Los* (RRL) was provided by Jiuzhou Yongding Agricultural Co., Ltd. (Langfang, China). Neutral protease (50,000 U/g) from *Bacillus subtilis* fermentation was purchased at Beijing Biotopped Technology Co. (Beijing, China), and a Lowry protein assay kit was purchased from Beijing Solarbio Science & Technology Co., Ltd. (Beijing, China). The ·OH scavenging rate kit was purchased from Suzhou Grace Biotechnology Co., Ltd. (Suzhou, China). The HepG2 cells used in this study were purchased from the BeNa Culture Collection (Beijing, China). DMEM, FBS, and DMSO were purchased from GIBCO Co., Ltd. (Carlsbad, CA, USA). Cell counting kit-8 (CCK-8) and the BCA assay kit were obtained from Biorigin (Beijing) Inc. (Beijing, China). DPPH, ABTS, ROS assay kit, MDA assay kit, SOD assay kit with NBT, GSH-Px assay kit with NADPH, and the CAT assay kit were purchased from Beyotime Biotechnology (Beijing, China). H_2_O_2_ was purchased from Aladdin Trading Co., Ltd. (Shanghai, China).

### 2.2. Preparation of Hydrolyzed Peptides of RRL

RRL powder was produced by the process of drying and grinding, and RRL hydrolyzed peptides were obtained by enzymatic hydrolysis. A total of 35 g of RRL was added to 2 L of deionized water, and the pH was adjusted to 7.0. It was then added to 12.6 g of neutral protease and hydrolyzed for 175 min at 50 °C. Afterwards, it was inactivated in a water bath in boiling water for 10 min, cooled to room temperature, and centrifuged at 4 °C and 12,000 rpm for 15 min. The ultrafiltration equipment was subjected to ultrafiltration using a 10 kDa molecular weight filter membrane at room temperature, and the supernatant was taken and freeze-dried. The lyophilized powder of hydrolyzed peptides of RRL was stored at −20 °C, and the antioxidant properties were determined.

### 2.3. Measurement of Hydrolyzed Peptide Concentration

The hydrolyzed peptide concentration was measured using the two-step Lowry method [12]. In the first step, the hydrolyzed peptide was combined with Cu^2+^ under alkaline conditions for 10 min to form a complex. In the second step, the complex was allowed to react with Folin’s reagent at 37 °C for 30 min, resulting in a dark blue color. The hydrolyzed peptide content was then measured at 650 nm and calculated by plotting the standard curve through the BSA standard solution.

### 2.4. Measurement of ABTS Radical Scavenging Capacity

The determination of the free radical activity of ABTS was based on the study of Lee et al. [13], with some modifications. The ABTS solution was prepared in a 1: 1 (*v*/*v*) ratio of 7 mM ABTS and 2.45 mM potassium persulfate, and a solution of 5 mM PBS was mixed and left in the dark for 12–16 h. Before use, the ABTS free radical solution was kept sealed from light, and the initial absorbance at 734 nm was determined by diluting the stock solution with PBS to give an initial absorbance of 0.70 ± 0.02. Subsequently, 190 µL of ABTS radical solution was mixed with 10 µL of RRL that had different amounts of hydrolyzed peptides in it. This mixture was then kept at room temperature in the dark for 6 min. The absorbance values of the samples were recorded at 734 nm (A_s_). As the control (A_c_), a 5 mM PBS solution containing samples was used. As the blank (A_b_), include the ABTS radical solution with water. The ABTS radical scavenging activity was measured as follows:(1)$$ABTS\;radical\;scavenging\;activity\left(\%\right)=\frac{A_{b}-A_{s}+A_{c}}{A_{b}}\times 100$$

### 2.5. Measurement of DPPH Radical Scavenging Capacity

The scavenging capacity of DPPH radicals was measured using the method of Tao et al. [14], with minor modifications. A 0.1 mmol/L ethanolic solution of DPPH was prepared by dissolving 10 mg of DPPH in 25 mL of anhydrous ethanol. The mixture was then ten times diluted.

The solution was sealed and stored away from light. The RRL hydrolyzed peptide (50 μL) was mixed with the DPPH radical solution (150 μL) and kept out of the light for 30 min. The absorbance of the sample was recorded at 517 nm (A_s_). The samples were treated with anhydrous ethanol as a control (A_c_), and the samples were treated as a blank (A_b_) with a DPPH radical solution in water. The DPPH radical scavenging activity was measured as follows:(2)$$DPPH\;radical\;scavenging\;activity\left(\%\right)=\frac{A_{b}-A_{s}+A_{c}}{A_{b}}\times 100$$

### 2.6. Determination of Hydroxyl Radical (·OH) Scavenging Capacity

The scavenging ability of hydroxyl radicals in hydrolyzed peptides of RRL was measured by the kit. Based on the principle that the amount of H_2_O_2_ is proportional to the ·OH produced by the Fenton reaction, the ability of the samples to scavenge hydroxyl radicals can be assessed by measuring the product of the reaction between ·OH and salicylic acid at 510 nm and determining the absorbance magnitude of the samples. The samples and reagents were mixed according to the requirements, and then reacted at 37 °C for 20 min. Then, 150 μL of the clarified solution was transferred into a 96-well plate, and the absorbance value (A_s_) was measured immediately at 510 nm. Blank values were obtained using water and reagents (A_b_), and the control absorbance value was obtained (A_c_). The scavenging activity of ·OH radicals was measured as follows:(3)$$\cdot OH\;radical\;scavenging\;activity\left(\%\right)=\frac{A_{b}-A_{s}+A_{c}}{A_{b}}\times 100$$

### 2.7. Ultrafiltration

The hydrolyzed peptides of Ultrafiltration RRL were separated using the ultrafiltration apparatus. The hydrolyzed peptides were diluted with deionized water and separated by a 3 kDa and 1 kDa ultrafilter membrane at 0.2 MPa pressure at room temperature. Three fractions of S1 (3–10 kDa), S2 (1–3 kDa), and S3 (<1 kDa), respectively, were finally obtained, which were lyophilized and stored at −20 °C, and their antioxidant activities were determined. 

### 2.8. Gel Filtration Chromatography

The S3 solution (60 mg/mL) was further isolated and purified by elution on a Sephadex G-15 column using ultrapure water with the method of Li et al. [15]. Elution was carried out with the same solution at 0.5 mL/min for 400 min, and the absorbance was measured at 220 nm to get an elution curve. After lyophilization, the antioxidative ability was measured. 

### 2.9. Peptide Sequence Analysis by LC-MS/MS

A sample of S3G1 was subjected to reductive alkylation and desalting using the method by Zhang et al. [16] and then analyzed by liquid mass spectrometry (LC-MS/MS) using an Easy-nLC 1200 liquid chromatograph (column 150 μm × 150 mm, C18, 1.9 μm, 100 Å) and Q Exactive™ Hybrid Quadrupole-Orbitrap™ mass spectrometer. The HPLC mobile phase A was composed of an aqueous solution of 0.1% formic acid, and the mobile phase B contained 0.1% formic acid and 80% acetonitrile at a flow rate of 600 nL/min. The mobile phase loadings were as follows: 0 min (4% B), 2 min (8% B), 45 min (28% B), 55 min (40% B), 56 min (95% B), and 66 min (95% B). Mass spectrometry was carried out with a scanning range of 100–1500 *m*/*z* and a resolution of 70,000. The maximum IT was 100 ms, and the AGC target was 3 × 10^6^. The peptide sequence analyses were obtained by retrieving and analyzing the raw files of the mass spectrometry acquisition files using PEAKS Studio 10.6 De novo software. Only short-chain peptides with high confidence and high content were chosen.

### 2.10. Computer Analysis of Identified Peptides

The Peptide Ranker server (http://distilldeep.ucd.ie/PeptideRanker/) (accessed on 13 June 2023) was used to estimate the potential biological activity of LC-MS/MS. Peptides with a Peptide Ranker score higher than 0.5 were considered to have potential biological activity, while those with higher bioactivity scores of 0.8 were chosen for the subsequent experiments. The toxicity of peptides was predicted by Toxin Pred (https://webs.iiitd.edu.in/raghava/toxinpred/multi_submit.php) (accessed on 21 June 2023), and chemical properties were determined by Innovagen (http://www.innovagen.com/proteomics-tools) (accessed on 2 July 2023). PEPTIDE 2.0 (https://www.peptide2.com/N_peptide_hydrophobicity_hydrophilicity.php) (accessed on 17 July 2023) was used to determine the percentage of hydrophobic amino acid residues. The peptides with 80% hydrophobic amino acids were then tested for allergenicity using AllerTOP 1.0 (http://www.ddg-pharmfac.net/allertop/index.html) (accessed on 21 July 2023). Eventually, the BIOPEP database (https://biochemia.uwm.edu.pl/biopep-uwm/) (accessed on 24 July 2023) was used to determine whether the peptides were novel peptides or not.

### 2.11. Molecular Docking

The P1 (LKPPF) and the P3 (LPFRP) peptides were studied using the molecular docking method [17]. The structures of the P1 and the P3 peptides were drawn using ChemDraw 22.0.0 software (PerkinElmer, Waltham, MA, USA). The Keap1 crystal structure (PDB ID: 2FLU) was downloaded as the receptor from the PDP database (https://www.rcsb.org/downloads) (accessed on 4 August 2023). The receptor was also manipulated using Pymol 2.6 software (San Carlos, CA, USA) to remove the water molecules and the Nrf2 16-mer fraction, then hydrogenated using AutoDockTools-1.5.7 (Computational Structural Biology, La Jolla, CA, USA), and energy was minimized using SPDBV 4.10 (Computational Structural Biology, La Jolla, CA, USA). The peptides were then hydrogenated as ligands using AutoDock VINA 1.1.2 software (Computational Structural Biology, La Jolla, CA, USA), through which the receptor and ligand were docked for analysis, where the Grid Box centers (17.717, 16.639, 7.056) and dimensions (57.05, 57.05, 57.05) were determined by selecting Vina docking to determine the docking site occupying the Nrf2 binding site and binding energies. The results were analyzed and processed visually using Pymol. The receptor and ligand binding sites and two-dimensional interaction maps were also examined using two online sites, PLIP (https://plip-tool.biotec.tu-dresden.de/plip-web/plip/index) (accessed on 11 September 2023) and Proteins Plus (https://proteins.plus/) (accessed on 12 September 2023).

### 2.12. Cell Culture and Cytotoxicity Assays

#### 2.12.1. Establishment of an H_2_O_2_-Induced Oxidative Stress Model in HepG2 Cells

The experimental design was adapted and refined by Xu et al. [18] and Zhao et al. [19]. In DMEM medium containing 10% FBS, HepG2 cells were cultured and incubated in a constant temperature incubator (37 °C, 5% CO_2_). The HepG2 cells were subcultured when 80–90% coverage was achieved. Cells in the logarithmic growth phase were seeded into separate well plates for subsequent experiments, with cell generations of 10–20 being suitable for the experiments.

Using He et al. [20], the oxidative stress model of HepG2 cells was measured for CCK-8. The 1 × 10^5^ cells/mL HepG2 cells were cultured for 24 h in 96 well plates, (6 replicate wells per group), and the blank group (A_o_), control group (A_c_), and experimental group (A_s_) were established, where the control group contained the cells, while the experimental group contained the H_2_O_2_ at concentrations of 250, 500, 750, 1000, 1250, 1500, 1750, and 2000 μM. After the cells continued to be incubated in the incubator for 24 h, the spent culture medium was discarded, and 100 μL of CCK-8 solution was added to each well to continue the incubation for 1.5 h to detect the OD value at 450 nm, and the experiment was repeated three times. The following formula was used: (4)$$Cell\;viability(\%)=\frac{A_{s}-A_{0}}{A_{c}-A_{0}}\times 100$$

#### 2.12.2. Effect of Novel Peptides on the Survival of HepG2 Cells

Method Section 2.12.1. was used to test the survival rate of HepG2 cells with new peptides. The experimental group included cells and peptide solutions at concentrations of 125, 250, 500, 1000, and 2000 μg/mL. Eventually, cell viability was measured using the CCK-8 method, and the experiment was performed three times.

#### 2.12.3. Effect of Novel Peptides on the Survival of Oxidatively Damaged HepG2 Cells by H_2_O_2_

HepG2 cells were grown in 96-well plates for 24 h using 100 μL at 1 × 10^5^ cells/mL. set up blank, control, model (1250 μM H_2_O_2_), and experimental (125, 250, 500, and 1000 μg/mL novel peptide + 1250 μM H_2_O_2_) groups were established to incubate the cells in the incubator. After additional incubation for 24 h, the spent culture media was discarded, and cell viability was measured using the procedure in Section 2.12.1, with the experiment repeated three times.

#### 2.12.4. Determination of ROS Levels in HepG2 Cells

Intracellular ROS were analyzed by digesting and collecting HepG2 cells from the control, model, and experimental groups in Section 2.12.3, precipitating the cells, and adding the DCFH-DA probe [21]. Following the steps in the ROS assay kit, the cells were treated with an excitation wavelength of 488 nm and an emission wavelength of 525 nm. The assay compared the fluorescence values found in the control group.

#### 2.12.5. Determination of MDA, SOD, GSH-Px, and CAT Activities in HepG2 Cells

The subsequent experiments were carried out according to the control, model, and experimental groups obtained in Section 2.12.3. Following the measurement of cellular protein concentrations using the BCA kit, cell lysates were used to produce protein solutions. In HepG2 cells, the MDA content, SOD, GSH-Px, and CAT activities were ascertained following the directions of the kit.

### 2.13. Statistical Analysis

Graphs were created using Origin 2021 software, statistical analysis was performed using SPSS 23.0 at the 5% level, and half-inhibitory concentration (IC_50_) values were calculated using GraphPad Prism 8.0. The data is shown as the mean ± SD, and each test was run three times in duplicate. A *p*-value of less than 0.05 was used to determine a statistically significant difference in the means. 

## 3. Results and Discussion 

### 3.1. Chemical Antioxidant Activity of Hydrolyzed Peptides of RRL

Antioxidant activity in vitro is most commonly assessed by the ability to scavenge free radicals. The ABTS radical is a common reagent for screening antioxidant radical scavenging activity, which assesses the relative ability of the antioxidant to scavenge ABTS from solution [22]. The DPPH radical, which accepts electrons or hydrogen from an antioxidant to form a stable product, is a commonly found free radical that assesses the antioxidant activity of the substance [23]. The rate of scavenging of the hydroxyl radical (·OH) is one of the important indicators of antioxidant activity, and it is one of the most active free radicals in the human body. It can immediately react with its neighboring organic and inorganic molecules, and may cause oxidative DNA damage, leading to damage to the organism, so the scavenging of hydroxyl radicals is of great significance in protecting the organism [24]. Therefore, the ABTS, DPPH, and hydroxyl radical scavenging capacities of RRL hydrolyzed peptides were determined.

The scavenging capacity of RRL hydrolyzed peptide against three free radicals is shown in Figure 1a–c, which showed a strong free radical scavenging capacity and a positive correlation with the hydrolyzed peptide concentration within the assay range, which is consistent with the previous report [25]. When the concentration of RRL was 500 μg/mL, the scavenging rate of ABTS radical and DPPH radical reached 84.82 ± 0.32% and 71.23 ± 2.03%, respectively, and when the concentration of hydrolyzed peptide was 5 mg/mL, the scavenging rate of hydroxyl radical reached 89.17 ± 2.37%, and its IC_50_ for the scavenging activities of ABTS, DPPH, and hydroxyl radicals were 174.7 ± 1.91 μg/mL, 260.3 ± 4.88 μg/mL, and 1.86 ± 0.07 mg/mL, respectively. 

Hu et al. [26] found that neutral protease had better in vitro antioxidant capacity against grateloupia livida protein than Protomax and alkaline protease (IC_50_ values of 0.88 ± 0.13 mg/mL for ABTS and 3.96 ± 0.41 mg/mL for DPPH), indicating that neutral protease may be better than the other two enzymes in terms of antioxidant capacity against hydrolyzed peptides. Feng et al. [27] determined the antioxidant properties of Chinese chestnut peptides with IC_50_ values of 15.35 ± 0.04, 18.21 ± 0.40, and 17.14 ± 0.33 mg/mL for ABTS, DPPH, and hydroxyl radical scavenging capacity, respectively. Ahmidin et al. [28] determined the IC_50_ values of 7.8 ± 0.09 mg/mL and 6.6 ± 0.03 mg/mL for ABTS and DPPH radical scavenging capacity of hydrolyzed peptides obtained by trypsinization of Bactrian camel milk. Rabiei et al. [29] investigated the antioxidant properties of Klunzinger’s mullet protein hydrolysate and determined the IC_50_ values for its scavenging activities against ABTS, DPPH, and hydroxyl radicals to be 0.60–0.12, 3.18–2.08, and 4.13–2.07 mg/mL, respectively. The higher the antioxidant capacity of the samples, the lower the corresponding IC_50_ values [30], indicating that the hydrolyzed peptides of RRL had a better scavenging activity for the three types of free radicals compared to other hydrolyzed peptide products; hence, the separation and purification of this component were carried out.

### 3.2. RRL Hydrolyzed Peptide Ultrafiltration Fractions

RRL hydrolyzed peptides were separated into S1 (3–10 kDa), S2 (1–3 kDa), and S3 (<1 kDa) components after ultrafiltration, and the scavenging activities of ABTS, DPPH, and hydroxyl radicals were compared with each component at the same peptide concentration, and the IC_50_ value of each component for the three free radicals was determined simultaneously.

The radical scavenging activities of the components after ultrafiltration of RRL hydrolyzed peptides are shown in Figure 2a–c. Free radical scavenging activities were seen across all three S1–S3 molecules with different molecular weights, and there was a positive correlation between the three assay concentrations and the amount of free radical scavenging activities, and S3 had a stronger free radical scavenging activity than the other two fractions. When the sample concentration was 500 μg/mL, the scavenging rates of S3 reached 86.82 ± 0.71% and 78.62 ± 4.04% for ABTS and DPPH radicals, respectively, and the scavenging rate of S3 reached 98.62 ± 1.08% for hydroxyl radicals since the level of the sample was 5 mg/mL, respectively. Figure 2d shows that compared to the other two ultrafiltration fractions and RRL hydrolyzed peptides, the S3 fraction had considerably lower IC_50_ values for ABTS, DPPH, and hydroxyl radical scavenging, at 157.3 ± 6.61 μg/mL, 190.6 ± 4.43 μg/mL, and 1.45 ± 0.03 mg/mL, respectively.

Tao et al. [14] determined the ABTS, DPPH, and hydroxyl radical scavenging activities of the hydrolyzed peptide fractions of Moringa leaf. The IC_50_ values for the <1 kDa sample were 0.83 mg/mL, 0.82 mg/mL, and 1.58 mg/mL, respectively, and the S3 fraction exhibited greater free radical scavenging activity when compared. Moreover, consistent with earlier research, many investigations have shown that ultrafiltration fractions with low molecular weights limit free radical scavenging activity more effectively [16,31,32]. In addition, research has demonstrated that fractions with a low molecular weight are quite effective at scavenging radicals, which helps in the reaction with radicals [33]. In summary, in comparison to the RRL hydrolyzed peptides and the other two fractions, the S3 fraction exhibited substantially lower IC_50_ values for ABTS radicals, DPPH radicals, and hydroxyl radicals, which proved that the S3 fraction had a stronger radical scavenging activity, and therefore, the S3 fraction was further investigated. 

### 3.3. Gel Filtration Chromatography of the S3 

Figure 3a shows that the two fractions, S3G1 and S3G2, were produced by further separation and purification on a Sephadex G-15 gel filtration column. The maximum antioxidant activity among the fractions was revealed by S3, which was obtained by ultrafiltration. The fractions were collected, evaporated, and lyophilized to determine their scavenging activities against ABTS and DPPH radicals. Component S3G1 displayed the most potent antioxidant activity, as demonstrated in Figure 3b,c. At 500 μg/mL, its scavenging rate against ABTS and DPPH radicals was 93.27 ± 1.84% and 88.96 ± 3.11%, respectively, with IC_50_ values of 153.0 ± 3.61 μg/mL and 170.2 ± 5.30 μg/mL. This indicated that S3G1 possessed the strongest antioxidant capacity and could potentially be isolated and identified to obtain the peptide sequences. 

### 3.4. Identification of Antioxidant Peptides by LC-MS/MS 

The molecular weight, hydrophobicity, amino acid content, and sequence of peptides are thought to correlate with their antioxidant properties [34]. The amino acid sequence and molecular weight of the purified antioxidant fraction (S3G1) were determined using LC-MS/MS [35]. De novo analysis was performed using PEAKS Studio 10.6 De novo software, and 121 peptides were isolated using the above method. The amino acid sequences, number of peptides, and other information are shown in Table 1.

Generally, antioxidant peptides from food sources are 2–15 amino acids in length, and the composition, number, and order of amino acids are significant factors influencing the antioxidant properties of peptides [36,37]. Peptides with hydrophobic amino acid residues at the N-terminal position may have strong antioxidant activity [38], and the aromatic amino acids add to the peptides’ antioxidant capacity [39]. It has been shown that hydrophobic amino acids are essential for the antioxidant properties of peptides [27,31].

The above peptides were predicted using a variety of software to select active peptides with high potential bioactivity, non-toxicity, good solubility, and a high percentage of hydrophobic amino acids, as shown in Table 2, as well as to determine whether the peptides were allergens and whether the peptides were reported. While the P2 (PFPPR) and the P4 (PPAPR) may be allergenic, P1 (LKPPF) and P3 (LPFRP) both have Leu at the N-terminal end, and the peptide chains contain hydrophobic amino acids as well as aromatic amino acids, such as Pro and Phe, and they may not be allergenic. Therefore, they may be desirable novel antioxidant peptides, whose secondary mass spectra are shown in Figure 4a,b, and whose molecular weights and chemical structures are shown in Figure 4c,d.

### 3.5. Molecular Docking

A protein called Kelch-like ECH-associated protein 1 (Keap1) can bind to nuclear factor erythroid-2 related factor 2 (Nrf2) and destroy it. Nrf2 is a key factor in the regulation of oxidative stress in cells and can be activated to regulate a variety of cellular antioxidant enzymes. By binding to the active site of Keap1, the bioactive peptide can interfere with the interaction between Nrf2 and Keap1, thereby reducing oxidative stress in human cells [40]. P1 and P3 resemble the antioxidant peptides found in walnuts that bind to Nrf2, suggesting that they may have non-covalent interactions with the Keap1 active site. Figure 5 shows the P1 structure (Figure 5a,c) and the P3 structure (Figure 5b,d), which reveal 3D and 2D interactions at the Keap1 active site, respectively [16]. As shown in Table 3, Keap1 forms six hydrophobic forces with P1, including three key residues (Tyr334, Tyr572, Phe577); eight hydrogen bonds, including four key residues (Arg380, Arg415, Gln530, Ser602); one π-π stacking, including one key residue (Tyr572); and one salt bridge, including one key residue (Arg415), while the docking score was −12.8 kcal/mol. Similarly, Table 4 shows (Arg415, Tyr525, Tyr572, Phe577) nine hydrogen bonds, including four key residues (Arg380, Asn382, Arg415, Arg483); one π-π stacking containing one key residue (Tyr572); and one salt bridge containing one key residue (Arg380). The docking score was −11.7 kcal/mol simultaneously. Zhang et al. performed molecular docking of two peptides, EYWNR and FQLPR, with Keap1, and showed that both peptides could generate hydrophobic force, hydrogen bonding, and π-stacking with Keap1, and EYWNR could also generate a salt bridge with Keap1. The two peptides, EYWNR and FQLPR, contained hydrophobic interactions with Tyr525 of Keap1-Kelch and hydrogen bonds with Arg380 and Ser555. On the other hand, both RLL P1 and P3 peptides can interact with Keap1 in all four forcings, including hydrophobic interaction with Tyr572 and Phe577 of RLL, hydrogen bonding with Arg380 and Arg415, and π-bonding with Tyr572. Hu et al. [26] molecularly docked four Grateloupia livida polypeptides with Keap1, and the binding sites occupied by the four polypeptides to form hydrogen bonds with the Keap1-Kelch region were all different to a certain extent, with the main binding sites being Arg380, Arg415, Arg483, Asn382, and Tyr334. The main binding sites of the P1 and P3 peptides were Arg380 and Arg415, which proved that the binding sites were relatively similar. P1 and P3 may be able to connect to Nrf2 in the Keap1-Kelch structural domain, according to analysis of the binding sites of the interacting forces. This would allow the free Nrf2 to be released. Thus, it is possible that the antioxidant peptides included in RRL effectively decrease the organism’s oxidative stress. 

### 3.6. Effect of Novel Peptides on the Survival of Oxidatively Damaged HepG2 Cells by H_2_O_2_

Modeling oxidative stress damage in hepatocellular carcinoma cells frequently involves using a common ROS with a non-radical form, H_2_O_2_, which can cross the cell membrane to damage the cells [41]. H_2_O_2_ was therefore chosen to induce cell damage. The protective effect of the P1 and P3 treatment on normal HepG2 cells was investigated to determine the effect of two novel peptides, P1 and P3, on the survival of H_2_O_2_-induced HepG2 cells. 

As seen in Figure 6a, the CCK-8 assay was used to assess the viability of HepG2 cells that had been stimulated by H_2_O_2_. In the concentration range of 500–2000 μmol/L of H_2_O_2_, the cell viability of HepG2 cells was discernibly reduced (*p* < 0.05), with the decrease in cell viability being an essential marker for the successful establishment of the oxidative damage model, in which the IC_50_ value of H_2_O_2_ was 1180 μmol/L. Therefore, significant damage to HepG2 cells was more likely to be caused by high concentrations of H_2_O_2_ than by lower concentrations, and according to the above results, 1250 μmol/L was set as the concentration of H_2_O_2_ for the subsequent experiments. Figure 6b demonstrated that peptides of P1 and P3 both enhanced cell growth within the 125–1000 μg/mL concentration range. However, the number of cells in the treated group was lower than in the control group when the peptide concentration reached 2000 μg/mL, so in the following experiments, the peptides were used within the 125–1000 μg/mL concentration range.

Figure 6c shows that the two peptides reduced the mortality rate of HepG2 cells that had been damaged by H_2_O_2_ oxidation. H_2_O_2_ considerably reduced the cell survival rate in the model group compared to the control group (*p* < 0.05), proving that the damage model was effectively set up. In contrast, the experimental group was treated with 1250 μmol/L H_2_O_2_ after the peptides were used to protect normal HepG2 cells, which significantly improved the survival of HepG2 cells after the intervention by making the concentration of the P1 and P3 peptides 125–1000 μg/mL (*p* < 0.05). It was demonstrated that both peptides improved the H_2_O_2_-induced decrease in cell viability after pretreatment of HepG2 cells, while the P1 peptide was more effective in improving HepG2 cell viability compared to that of P3 at the same concentration.

### 3.7. Effect of Novel Peptides on ROS Levels in H_2_O_2_-Oxidatively Damaged HepG2 Cells

The mitochondrial electron transport cycle produces ROS, which the antioxidant system then scavenges; normally, ROS can maintain a balance between production and scavenging, but in the presence of external stimuli, ROS can accumulate and further damage cells, tissues, and even organs [42]. Two new peptides, P1 and P3, were tested for their impact on ROS levels in HepG2 cells that had been oxidatively damaged by H_2_O_2_ [43]. Antioxidant peptides are known to have a great capacity to scavenge ROS. The model group had a considerably higher amount of reactive oxygen species (ROS) than the control group (*p* < 0.05), as shown in Figure 6d. On the other hand, when the concentration of the two peptides increased, the relative fluorescence intensity of ROS in the experimental group dropped progressively. At concentrations ranging from 250 to 1000 μg/mL, the relative fluorescence intensity of the P1 peptide was less than that of the P3 peptide, and it reached (112.86 ± 3.83) RFU at the 1000 μg/mL concentration, which proved that the P1 peptide was more capable of reducing the ROS level compared with the P3 peptide.

### 3.8. Effect of Novel Peptides on MDA Contents in H_2_O_2_-Oxidatively Damaged HepG2 cells

ROS can react with unsaturated fatty acids to form lipid hydroperoxides, while MDA is an important metabolite of cellular lipid oxidation. Furthermore, MDA contents are higher in diseases caused by oxidative stress, making MDA contents an important indicator substance of cellular oxidative stress damage [44]. The effect of peptides on intracellular MDA content is shown in Figure 7a, which was significantly increased in the model group compared to the control group (*p* < 0.05). After being treated with P1 or P3, the experimental group was able to considerably lower the MDA level at values of 500 and 1000 μg/mL. Among them, the MDA content in HepG2 cells was (17.39 ± 0.36) μmol/mg pro at 1000 μg/mL for the P1 peptide, with an inhibition rate of 19.86%, while its inhibition rate was higher than that of the P3 at the same concentration (12.91%). In conclusion, there was a significant reduction in H_2_O_2_-induced MDA content in HepG2 cells by both peptides at high concentrations (500 and 1000 μg/mL), while the inhibition rate of the P1 peptide was higher than that of the P3.

### 3.9. Effect of Novel Peptides on Antioxidant Enzymes in H_2_O_2_-Oxidatively Damaged HepG2 Cells

A crucial defense mechanism against oxidative stress is the intracellular antioxidant enzyme system. Enzymes like SOD, GSH-Px, and CAT can lower free radicals and halt oxidative processes; an increase in antioxidant enzyme activity may lead to an increase in intracellular antioxidant capacity [45]. Hence, the SOD, GSH-Px, and CAT activities of HepG2 cells were examined to better understand how P1 and P3 protect HepG2 cells from H_2_O_2_ oxidative damage.

Figure 7b–d show that when cells were exposed to 1250 μmol/L H_2_O_2_, there was a notable drop (*p* < 0.05) in the activities of SOD, GSH-Px, and CAT. This suggested that the cells’ oxidative stress resulted in a reduction in antioxidant enzyme activities following H_2_O_2_ treatment. On the contrary, after pretreatment with different concentrations of peptides P1 and P3, the activities of SOD, GSH-Px, and CAT were increased compared to the model group. This result is consistent with the findings of Xu et al. [42], who found that the intracellular activities of three antioxidant enzymes were significantly increased in the monkey mushroom peptide KSPLY-pretreated group compared to the damaged group. This is also consistent with the results of peptides from Coix lacryma [46], wheat [47], and Moringa oleifera leaves [14]. The activities of the three antioxidant enzymes were increased by 93.65%, 31.69%, and 58.26% in the 1000 μg/mL P1 group, and 69.48%, 68.64%, and 51.43% in the P3 group, respectively, so that the activities of SOD, GSH-Px, and CAT were significantly improved by 1000 μg/mL of P1 and P3 compared with the model group (*p* < 0.01). The enzyme activities of SOD and CAT were elevated higher in P1 than in P3 at 1000 μg/mL, whereas the GSH-Px enzyme activities were elevated higher in P3. The two peptides were demonstrated to reduce antioxidant enzyme damage, possibly by scavenging some of the ROS and MDA, which promotes the functioning of antioxidant enzymes and reduces cellular damage from oxidative stress.

## 4. Conclusions

*Rumexpatientia L. ×Rumextianshanicus A. Los* has a high yield, a long lifespan, high adaptability, and wide distribution; however, the functionality of RRL peptides has rarely been reported. In the previous study, an industrial-grade neutral protease was selected, which has high catalytic efficiency and specificity and can not only catalyze protein hydrolysis quickly and efficiently, but also selectively cut specific peptide bonds of proteins. Moreover, the mild conditions of action are conducive to the protection of the natural structure and activity of the peptide, which can reduce the generation of industrial waste in later production, meeting the trend of modern industrial requirements for environmental protection. Subsequently, two novel peptides of RRL were identified using conventional antioxidant in vitro indexes assessed by gradient separation and peptide identification, screening, and molecular docking methods, which were determined to have protective effects against oxidative stress by cellular experiments. The hydrolyzed peptides were isolated, and the most potent antioxidant component was identified as S3G1, while the two peptides identified were LKPPF and LPFRP. The molecular docking studies showed that both peptides, LKPPF and LPFRP, interacted with the Keap1-Kelch domain and promoted Nrf2 release, while the two peptides were protective against oxidative damage in cells. Therefore, this study provides a theoretical basis for the hydrolyzed peptides of RRL, as well as the LKPPF and LPFRP antioxidant effects in RRL. Antioxidant peptides are a group of small-molecule peptides with antioxidant activity that scavenge free radicals, inhibit oxidative stress, and protect cells from oxidative damage. Whereas oxidative stress is associated with a variety of chronic diseases such as cardiovascular disease, diabetes, cancer, and neurodegenerative diseases, antioxidant peptides can help reduce the risk of these diseases and may have specific health-promoting or disease-preventing properties [48]. Compared to traditional synthetic antioxidants, antioxidant peptides are usually derived from natural sources, which are considered safer and healthier [49]. Antioxidant peptides typically have good thermal and pH stability, which allows them to be used under a wide range of food processing conditions without losing their activity. Therefore, RRL peptides, as well as LKPPF and LPFRP, may be further developed as food additives or functional food supplements. 

However, the peptides may be specifically recognized by various proteases during gastrointestinal digestion for specific amino acid sequences in the peptide chain, which catalyzes their transformation into small-molecule peptides or individual amino acids [50]. Gastrointestinal tract simulation, which mimics the digestive environment in the human body through the use of artificially simulated gastric and intestinal fluids containing the appropriate enzymes, is essential for evaluating the functionality, stability, and bioavailability of peptides in the gastrointestinal tract. Wang et al. [51] found from gastrointestinal tolerance studies that YDEAGPSIVH had greater gastrointestinal tolerance but the obtained fragments were not biologically active, whereas the gastrointestinal stability of FAGDDAPRAVF was not as good as that of YDEAGPSIVH, but the gastrointestinal digestion may positively affect the α-glucosidase inhibitory activity of FAGDDAPRAVF. In summary, the subsequent studies with the simulation of the gastrointestinal tract are essential to further understand the digestive kinetics of peptides and to predict their bioactivity, contributing to the development of new functional food ingredients that guide the design and application of nutritional supplements. Currently, only the ROS, MDA, and antioxidant enzyme activities of the two peptides are measured, while the cellular antioxidant mechanism and pathway exploration are not involved. Meanwhile, animal models are able to mimic human physiology and pathology, as well as long-term safety and potential side effects [52]. The final clinical studies are a crucial stage in assessing the genuine effectiveness and safety of peptides in humans, which can objectively evaluate the antioxidant effects of peptides in humans and whether they can improve the symptoms of related diseases [53]. Therefore, additional studies are required to further investigate the mechanism of the antioxidant property of its peptide and its other functional studies in the human body through cellular pathways, animal models, and clinical studies.

## Figures and Tables

**Figure 1 foods-13-00981-f001:**
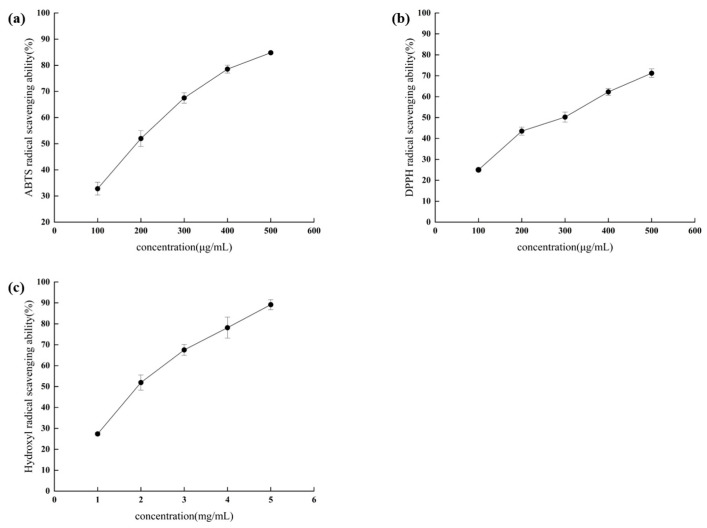
Scavenging activities of RRL hydrolyzed peptides against ABTS radicals (**a**), DPPH radicals (**b**), and hydroxyl radicals (**c**).

**Figure 2 foods-13-00981-f002:**
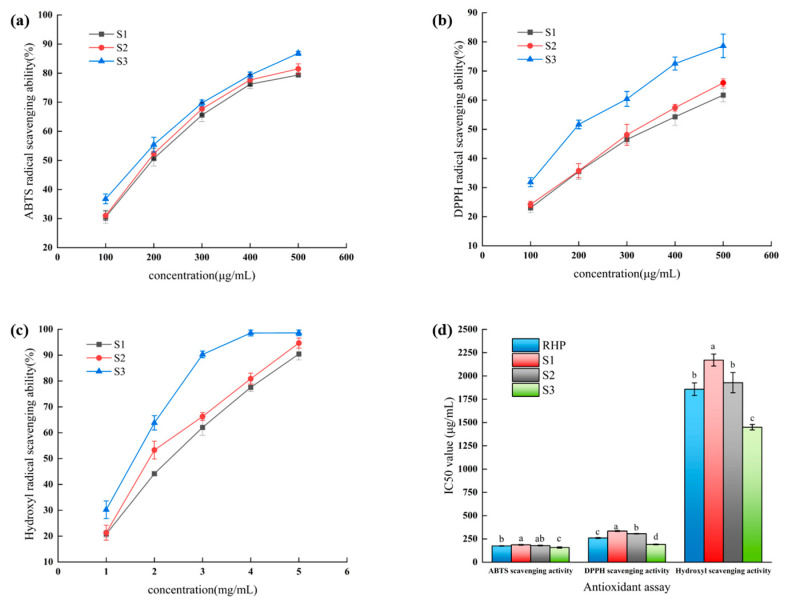
Scavenging activity of RRL hydrolyzed peptide ultrafiltration fractions against ABTS radicals (**a**), DPPH radicals (**b**), hydroxyl radicals (**c**), and IC_50_ of the different fractions (**d**). Error lines indicate the standard deviation of the three measurements. (*p* < 0.05). (RHP: RRL hydrolyzed peptide; S1: 3–10 kDa peptide fractions; S2: 1–3 kDa peptide fractions; S3: <1 kDa). The different lowercase letters at the top of the pattern bar in the picture represent significant differences between groups.

**Figure 3 foods-13-00981-f003:**
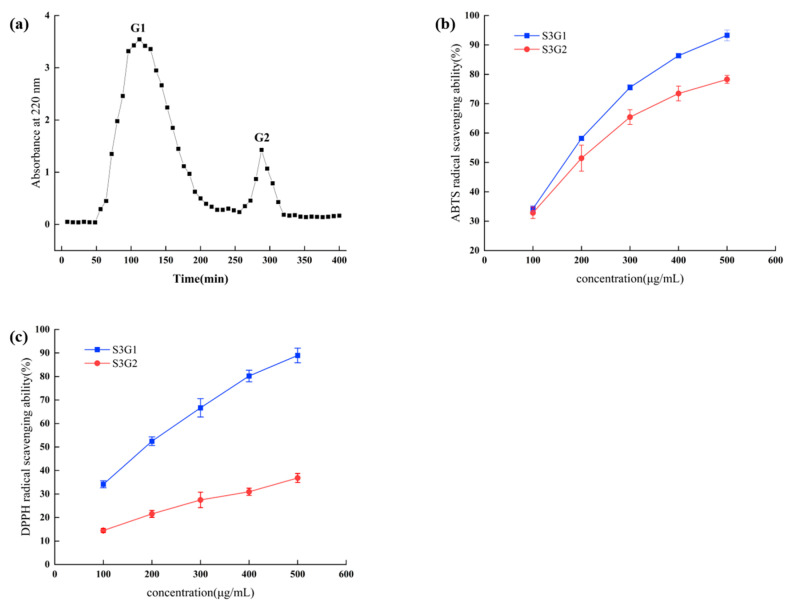
Chromatogram of a Sephadex G-15 gel filtration column for the separation of S3 (**a**) and the scavenging activities of ABTS radicals (**b**) and DPPH radicals (**c**) for the elution peaks. The error line indicates the standard deviation of the three measurements.

**Figure 4 foods-13-00981-f004:**
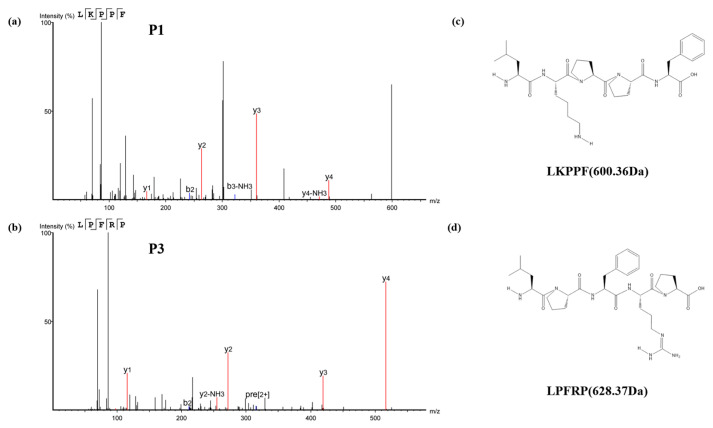
Mass spectra and chemical structures of the peptides P1 (**a**,**c**) and P3 (**b**,**d**). P1, LKPPF; P3, LPFRP.

**Figure 5 foods-13-00981-f005:**
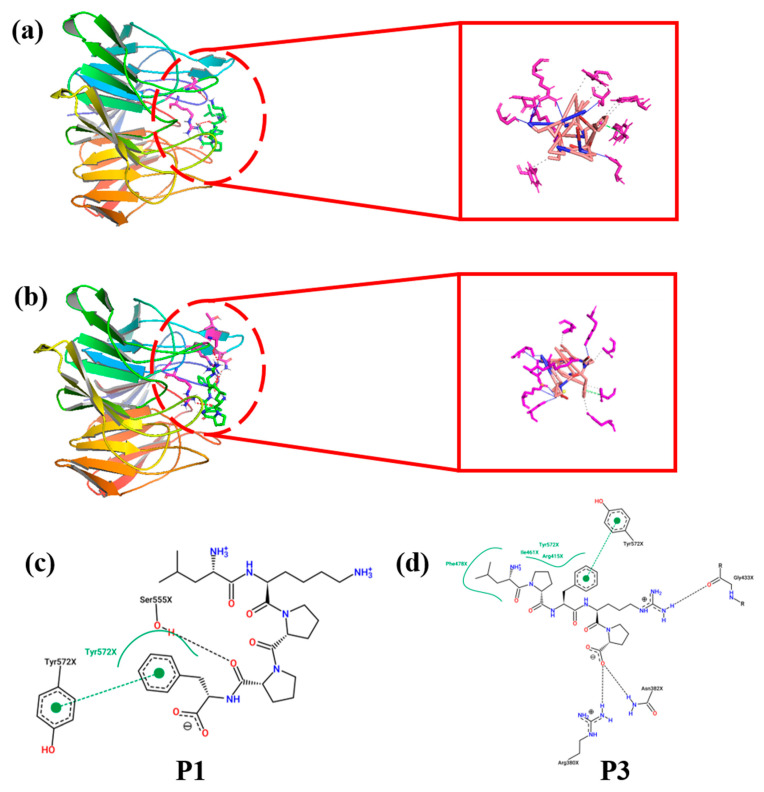
The 3D and 2D interactions of P1 (**a**,**c**) and P3 (**b**,**d**) with the Keap1 active site. P1, LKPPF; P3, LPFRP. (PDB:2FLU).

**Figure 6 foods-13-00981-f006:**
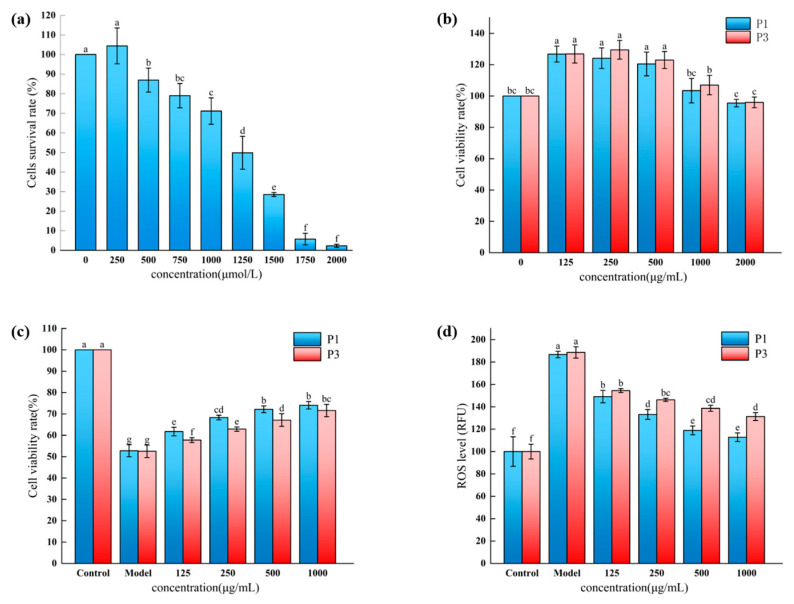
Effect of peptides P1 and P3 on cell viability in the presence of H_2_O_2_. (**a**) Effect of different concentrations of H_2_O_2_ on cell survival. (**b**) Effect of different concentrations of P1 and P3 on cell viability. (**c**) Protective effect of different concentrations of P1 and P3 from H_2_O_2_ damage. (**d**) Effect of different concentrations of P1 and P3 on ROS level. The different lowercase letters at the top of the pattern bar in the picture represent significant differences between groups. (P1, LKPPF; P3, LPFRP).

**Figure 7 foods-13-00981-f007:**
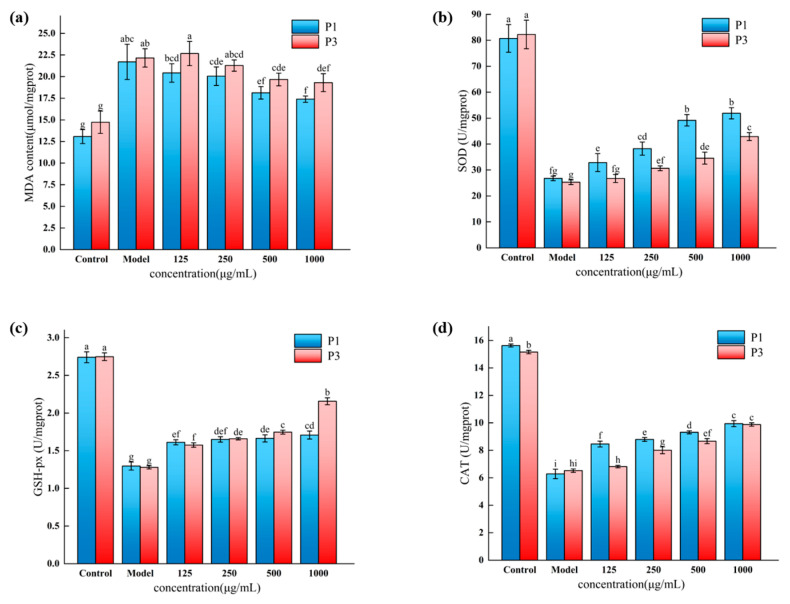
Effect of peptides P1 and P3 on MDA contents and antioxidant enzymes in the presence of H_2_O_2_. (**a**) Effect of different concentrations of P1 and P3 on MDA contents. (**b**) Effect of different concentrations of P1 and P3 on SOD enzyme. (**c**) Effect of different concentrations of P1 and P3 on GSH-Px enzyme. (**d**) Effect of different concentrations of P1 and P3 on CAT enzyme. The different lowercase letters at the top of the pattern bar in the picture represent significant differences between groups. (P1, LKPPF; P3, LPFRP).

**Table 1 foods-13-00981-t001:** Bioinformatic analysis of the peptides identified from S3G1.

No.	Peptide	Score	Length	RT	Area	No.	Peptide	Score	Length	RT	Area
1	LAYKPPR	99	7	11.12	5.22 × 10^8^	62	YKPPR	97	5	6.48	3.10 × 10^7^
2	AFDEGPWPR	99	9	32.62	2.39 × 10^8^	63	TDYPPLGR	97	8	19.66	2.87 × 10^7^
3	FAPSLPEKN	99	9	21.06	1.06 × 10^8^	64	LLYDDR	97	6	18.22	2.72 × 10^7^
4	LSDPWHNT	99	8	25.05	1.05 × 10^8^	65	LGRLSP	97	6	14.36	2.53 × 10^7^
5	LFEEPVPGK	99	9	23.61	1.00 × 10^8^	66	LKVPL	97	5	26.99	1.95 × 10^7^
6	TVLLPR	99	6	20.3	9.59 × 10^7^	67	LGNLRP	97	6	13.4	1.68 × 10^7^
7	RPLVMH	99	6	23.22	6.60 × 10^7^	68	LPKVP	97	5	20.07	1.58 × 10^7^
8	FTGSNVKVA	99	9	15.72	6.50 × 10^7^	69	LQVER	97	5	9	1.51 × 10^7^
9	AFRVP	99	5	23.53	3.99 × 10^7^	70	FVRLLG	97	6	27.79	1.48 × 10^7^
10	VVRLP	99	5	19.82	2.40 × 10^7^	71	LKAPA	97	5	7.49	1.46 × 10^7^
11	LLPR	99	4	10.81	2.25 × 10^7^	72	LGNNPAKGGL	97	10	14.67	1.34 × 10^7^
12	LLSPDPATK	99	9	15.21	1.77 × 10^7^	73	PFPPR	97	5	18.32	1.30 × 10^7^
13	VSPLEVK	99	7	16	1.35 × 10^7^	74	LFPRDPY	97	7	28.53	1.29 × 10^7^
14	LGDAFYYGK	99	9	29.33	1.30 × 10^7^	75	LVTGKGPLEN	97	10	14.07	1.28 × 10^7^
15	FLDDVQVK	99	8	28.11	1.31 × 10^7^	76	PPAPR	97	5	12.07	1.24 × 10^7^
16	PLLRP	98	5	15.62	6.32 × 10^8^	77	LLLPR	97	5	23.17	1.02 × 10^7^
17	LWYGPDRP	98	8	29.01	5.23 × 10^8^	78	LLKFE	97	5	24.44	1.01 × 10^7^
18	VLLPR	98	5	17.49	3.26 × 10^8^	79	WVPPEGK	96	7	18.19	1.33 × 10^9^
19	LGKVYDY	98	7	21.82	2.96 × 10^8^	80	LVVLGH	96	6	17.87	5.31 × 10^8^
20	VTLPR	98	5	14.51	1.43 × 10^8^	81	WYGPDRP	96	7	22.97	1.87 × 10^8^
21	FDEGPWRP	98	8	30.25	1.11 × 10^8^	82	SFRVTP	96	6	22.05	8.02 × 10^7^
22	LTPSSSFKDA	98	10	19.18	9.06 × 10^7^	83	WVPPEGR	96	7	19.66	7.20 × 10^7^
23	LRLP	98	4	22.05	7.11 × 10^7^	84	LKFE	96	4	15.82	5.08 × 10^7^
24	FSEYPPLGR	98	9	28.87	6.19 × 10^7^	85	VTGKGPFDNL	96	10	31.92	4.63 × 10^7^
25	FRVP	98	4	21.67	5.86 × 10^7^	86	LLDLH	96	5	25.81	3.91 × 10^7^
26	LGTVPVGR	98	8	15.21	5.70 × 10^7^	87	LLEGLPK	96	7	24.37	3.81 × 10^7^
27	FVPGK	98	5	8.5	5.05 × 10^7^	88	LFQGPPGH	96	8	18.48	2.59 × 10^7^
28	LTKLGVK	98	7	9	4.76 × 10^7^	89	LGDAFYYNK	96	9	29.07	2.51 × 10^7^
29	LFPR	98	4	15.62	4.53 × 10^7^	90	LLKVP	96	5	21.03	2.33 × 10^7^
30	LRVP	98	4	16.35	4.53 × 10^7^	91	TFSEYRGLPP	96	10	30.99	2.29 × 10^7^
31	FRLP	98	4	25.11	4.30 × 10^7^	92	LVMHDY	96	6	18.07	2.22 × 10^7^
32	FTGKQPYDL	98	9	27.35	3.52 × 10^7^	93	LLKFDP	96	6	30.99	1.89 × 10^7^
33	WYGPDR	98	6	18.22	3.08 × 10^7^	94	VVRLPYD	96	7	27.02	1.75 × 10^7^
34	YKPP	98	4	9.41	2.91 × 10^7^	95	LLTDFKP	96	7	26.67	1.65 × 10^7^
35	AFRVTP	98	6	22.05	2.69 × 10^7^	96	LELPR	96	5	19.56	1.60 × 10^7^
36	LTPSSSFK	98	8	15.21	2.53 × 10^7^	97	LLPVGR	96	6	15.85	1.53 × 10^7^
37	VVRLYP	98	6	28.18	2.44 × 10^7^	98	LLGFDNVRQ	96	9	30.64	1.48 × 10^7^
38	VKTPW	98	5	22.37	2.42 × 10^7^	99	LDPVLGR	96	7	19.54	1.35 × 10^7^
39	LTPSSSFKD	98	9	16.96	2.20 × 10^7^	100	YYDGR	96	5	8.47	1.19 × 10^7^
40	LNLSSESGKY	98	10	21.89	2.19 × 10^7^	101	LPRPRP	96	6	8.99	1.19 × 10^7^
41	LPWKD	98	5	22.05	1.89 × 10^7^	102	AAKWSPE	96	7	14.58	1.17 × 10^7^
42	LSDPTHLGSP	98	10	20.45	1.74 × 10^7^	103	LKPPF	96	5	27.28	1.11 × 10^7^
43	FVPGR	98	5	10.43	1.56 × 10^7^	104	LHEDVPHTP	96	9	13.28	1.09 × 10^7^
44	LKAAP	98	5	7.49	1.46 × 10^7^	105	LKFP	95	4	24.6	9.92 × 10^7^
45	LDVLPHPQ	98	8	28.14	1.24 × 10^7^	106	VGYPGKY	95	7	17.33	5.27 × 10^7^
46	LFKDPF	98	6	36.75	1.18 × 10^7^	107	YYYDGR	95	6	15.85	4.30 × 10^7^
47	LRPP	98	4	10.59	1.10 × 10^7^	108	LLPH	95	4	11.03	3.75 × 10^7^
48	LLFGEK	98	6	25.68	1.02 × 10^7^	109	LLPPVDLK	95	8	29.33	3.29 × 10^7^
49	LKFPD	97	5	23.26	1.86 × 10^8^	110	LLLNR	95	5	15.47	2.92 × 10^7^
50	VTLRP	97	5	14.51	1.43 × 10^8^	111	HHSPGYYDGR	95	10	6.92	2.44 × 10^7^
51	LWYGDPR	97	7	25.11	1.34 × 10^8^	112	LNREPPEYRP	95	10	15.85	2.05 × 10^7^
52	YTPDYEPH	97	8	17.72	1.27 × 10^8^	113	KFPERAGP	95	8	10.43	1.96 × 10^7^
53	LSNPTKY	97	7	14.83	1.10 × 10^8^	114	VTLPPR	95	6	12.87	1.96 × 10^7^
54	LPFRP	97	5	22.28	9.76 × 10^7^	115	LLLRP	95	5	22.27	1.61 × 10^7^
55	LLLGDR	97	6	18.45	8.40 × 10^7^	116	LRFGP	95	5	22.02	1.55 × 10^7^
56	STLPGNKY	97	8	14.64	8.22 × 10^7^	117	RPLVMHDY	95	8	32.3	1.48 × 10^7^
57	LVDPKGPF	97	8	27.7	7.71 × 10^7^	118	PGYPGKY	95	7	16.16	1.45 × 10^7^
58	LVDPKGFP	97	8	27.7	7.71 × 10^7^	119	WYGPDRKP	95	8	15.21	1.33 × 10^7^
59	LDGLPPAPR	97	9	22.65	6.59 × 10^7^	120	LLDALPK	95	7	26.48	1.25 × 10^7^
60	LLHF	97	4	28.27	4.91 × 10^7^	121	PLRPRP	95	6	9	1.19 × 10^7^
61	FTGKQPYD	97	8	13.44	3.67 × 10^7^						

**Table 2 foods-13-00981-t002:** Biological activity and physicochemical properties of selected peptides.

Fraction No.	Sequence	Peptide Ranker Score	Toxin	Estimated Solubility	Hydrophobic	Allergen	Peptide
P1	LKPPF	0.898368	Non-Toxin	Good	80.00%	Probable non-allergen	Novel peptide
P2	PFPPR	0.963098	Non-Toxin	Good	80.00%	Probable allergen	Novel peptide
P3	LPFRP	0.941112	Non-Toxin	Good	80.00%	Probable non-allergen	Novel peptide
P4	PPAPR	0.803504	Non-Toxin	Good	80.00%	Probable allergen	Novel peptide

**Table 3 foods-13-00981-t003:** Action sites of P1 (LKPPF) and Keap1 interaction force.

Intermolecular Forces	No.	Residue	AA
Hydrophobic Interactions	1	334X	Tyr
	2	334X	Tyr
	3	478X	Phe
	4	572X	Tyr
	5	577X	Phe
	6	577X	Phe
Hydrogen Bonds	1	380X	Arg
	2	380X	Arg
	3	414X	Asn
	4	415X	Arg
	5	431X	Ser
	6	431X	Ser
	7	530X	Gln
	8	602X	Ser
π-Stacking	1	572X	Tyr
Salt Bridges	1	415X	Arg

**Table 4 foods-13-00981-t004:** Action sites of the P3 (LPFRP) and Keap1 interaction force.

Intermolecular Forces	No.	Residue	AA
Hydrophobic Interactions	1	415X	Arg
	2	478X	Phe
	3	525X	Tyr
	4	572X	Tyr
	5	572X	Tyr
	6	577X	Phe
	7	577X	Phe
Hydrogen Bonds	1	380X	Arg
	2	380X	Arg
	3	380X	Arg
	4	382X	Asn
	5	414X	Asn
	6	415X	Arg
	7	431X	Ser
	8	433X	Gly
	9	483X	Arg
π-Stacking	1	572X	Tyr
Salt Bridges	1	380X	Arg

## Data Availability

The original contributions presented in the study are included in the article, further inquiries can be directed to the corresponding author.

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
