# Peer review of "Preparation, Isolation and Antioxidant Function of Peptides from a New Resource of Rumexpatientia L. ×Rumextianshanicus A. Los"

_foods, 2024, doi:10.3390/foods13070981_

Round 1
Reviewer 1 Report
Comments and Suggestions for Authors
The document "Preparation, isolation, synthesis and antioxidant function of 2 plant peptides from a new resource of Rumexpatientia L. ×Ru-3 mextianshanicus A. Los" contains numerous methodological and conceptual errors, making the article confusing. Additionally, it lacks organization and balance in the analysis and discussion of results.
The phrasing in lines 83-88 is unclear. Review the wording.
In some lines, the authors use the term peptide when they should mention peptides or hydrolyzed peptides (lines 93-94, 117).
Throughout the document, the units to refer to peptides should be kDa instead of KD.
Is the correct flow rate mentioned in line 155?
Why do the authors in section 2.12.3 use the term "Novel peptides" when the identified peptides have already been reported and are in a database?
The values mentioned in lines 307-308 do not match those in Figure 2.
Lines 329-331. Why is it stated as a single peptide? There could be several species of peptides, each with a weight less than 1 kDa.
There is an imbalance between the presented discussion and the obtained results. I suggest that the authors conduct a more profound analysis and merge the results and discussion sections.
Comments on the Quality of English Language
I suggest having the style and wording reviewed by a native English speaker for better clarity and coherence.
Author Response
Dear Editors and Reviewers:
Thank you for your letter and for the reviewers’ comments concerning our manuscript entitled “Preparation, isolation and antioxidant function of peptides from a new resource of Rumexpatientia L. x Rumextianshanicus A. Los” (ID: foods-2901464). The comments are all valuable and very helpful for revising and improving our paper, as well as the important guiding significance to our research. We have studied comments carefully and have made correction which we hope meet with approval.
Response to Reviewer 1 Comments
Reviewer #1: The document "Preparation, isolation, synthesis and antioxidant function of 2 plant peptides from a new resource of Rumexpatientia L. ×Rumextianshanicus A. Los" contains numerous methodological and conceptual errors, making the article confusing. Additionally, it lacks organization and balance in the analysis and discussion of results.
- The phrasing in lines 83-88 is unclear. Review the wording. (section 2.2 Preparation of Hydrolyzed Peptides of RRL)
The author's answer: We tried our best to improve the manuscript and made some changes to the manuscript. These changes will not influence the content and framework of the paper. And here we did not list the changes but marked in red in the revised paper. We appreciate for Editors/Reviewers’ warm work earnestly and hope that the correction will meet with approval.
- In some lines, the authors use the term peptide when they should mention peptides or hydrolyzed peptides (lines 93-94, 117).
The author's answer: We sincerely thank the reviewer for careful reading. As suggested by the reviewer, we have corrected the “peptide” into “hydrolyzed peptides”.
- Throughout the document, the units to refer to peptides should be kDa instead of KD.
The author's answer: We were really sorry for our careless mistakes. Thank you for your reminder.
- Is the correct flow rate mentioned in line 155?
The author's answer: We feel great thanks for your professional review work on our article. The flow rate of 600 nL/min mentioned in line 155 is the correct flow rate, citing the DOI number "https://doi.org/10.3390/foods12071554".
- Why do the authors in section 2.12.3 use the term "Novel peptides" when the identified peptides have already been reported and are in a database?
The author's answer: Thank you again for your positive comments and valuable suggestions to improve the quality of our manuscript. We again looked for the two peptides from the BIOPEP database (https://biochemia.uwm.edu.pl/biopep-uwm/ ) and they did not appear in the database. It can be assumed that LKPPF and LPFRP are novel peptides that we found in RRL.
- The values mentioned in lines 307-308 do not match those in Figure 2.
The author's answer: On behalf of all the contributing authors, I would like to express our sincere appreciations of your letter and reviewers’ constructive comments concerning our article. Lines 307-308 show the original text as "Tao et al. determined the ABTS, DPPH, and hydroxyl radical scavenging activities of the peptide fractions of Moringa leaf. The IC50 values for the <1kDa sample were 0.83 mg/mL, 0.82 mg/mL, and 1.58 mg/mL" describing the subject as a hydrolyzed peptide from Moringa leaves, not RRL content.
- Lines 329-331. Why is it stated as a single peptide? There could be several species of peptides, each with a weight less than 1 kDa.
The author's answer: We have re-written this part according to the Reviewer’s suggestion, replace it with "This indicated that S3G1 possessed the strongest antioxidant capacity and could potentially be isolated and identified to obtain the peptide sequences".
- There is an imbalance between the presented discussion and the obtained results. I suggest that the authors conduct a more profound analysis and merge the results and discussion sections.
The author's answer: We thank the reviewer for pointing this out. We have revised. We took the discussion section and analyzed it in more depth, but as we received conflicting advice from another reviewer, we decided to make the change they suggested, because merging the results with the discussion we thought we might not be able to express the subjective language better, and writing the discussion separately might better convey the main idea of the paper. We hope this was the right decision.
We tried our best to improve the manuscript and made some changes marked in red in revised paper which will not influence the content and framework of the paper. We appreciate for Editors/Reviewers’ warm work earnestly and hope the correction will meet with approval. Once again, thank you very much for your comments and suggestions.
Yours sincerely,
Chang Liu
11 March 2024
School of Food and Health
Beijing Technology and Business University
Beijing 100048, China
E-mail: lc15373689836@163.com
aijinma206@sina.com

Reviewer 2 Report
Comments and Suggestions for Authors
In this study, two new antioxidant peptides, LKPPF and LPFRP, were obtained from RRL (Rumex patientia L. x Rumex tianschanicus A.Los) and applied to H2O2-treated HepG2 cells to investigate their antioxidant properties.
At low concentrations, both peptides were not cytotoxic to HepG2 cells, reduced the production of intracellular ROS and MDA, and increased the enzymatic activities of SOD, GSH-Px, and CAT.
The potential applications of food-derived antioxidant peptides as additives, nutraceuticals and therapeutic agents have fuelled the current interest in discovering them from different plant sources.
The authors also mention the potential use of two peptides studied as food additives or supplements. Whether this is the purpose of finding antioxidant peptides in the RRL needs to be better explained and discussed.
The topic is interesting and timely, but the paper has some serious gaps.
The work is sloppily written. Some examples.
Remove from the title the words: synthesis of 2 plant peptides. The authors, in fact, identify the peptides and determine their antioxidant activity but do not report the synthesis of the peptides in the methods.
The name of the so called “protein grass” is spelled wrong in the title, in abstract, and in lines 27, 68 etc. (Rumexpatientia L. x Rumextianshanicus A. Los instead of Rumex patientia L. x Rumex tianschanicus A.Los).
Authors must specify what kind of neutral protease they use in materials and methods.
The authors say that in China the leaves, flowers, and seeds of certain Rumex plant species are consumed as food. For the hydrolysis of RRL proteins, however, the authors use a neutral protease and not a standard gastrointestinal digestion method. They should explain why they use a neutral protease (perhaps to prepare peptides for industrial purpose?).
Furthermore, the authors say that RRL, known as "protein grass “, in China, was recognized as a “new food ingredient” in 2021. So, is it an ingredient or a food?
Considering these observations, the entire discussion must be rewritten, better explaining the objectives that the authors set themselves and the prospects. What is known about the efficacy of these peptides or RRL in vivo? The impact of gastrointestinal digestion on the efficacy, stability, and biological availability of these peptides?
Figures 6 and 7 are difficult to read and difficult to understand. They need to be redone and their legends rewritten. For example, Fig. 6 legend: Figure 6. Effect of peptides P1 and P3 on cell viability in the presence of H2O2. a) Effect of different concentrations of H2O2 on cell survival. b) Effect of different concentrations of P1 and P3 on cell viability. c) Protective effect of different concentrations of P1 and P3 from H2O2 damage. d) Effect of different concentrations of P1 and P3 on ROS level.
Author Response
Dear Editors and Reviewers:
Thank you for your letter and for the reviewers’ comments concerning our manuscript entitled “Preparation, isolation and antioxidant function of peptides from a new resource of Rumexpatientia L. x Rumextianshanicus A. Los” (ID: foods-2901464). The comments are all valuable and very helpful for revising and improving our paper, as well as the important guiding significance to our research. We have studied comments carefully and have made correction which we hope meet with approval.
Response to Reviewer 2 Comments
Reviewer #2 In this study, two new antioxidant peptides, LKPPF and LPFRP, were obtained from RRL (Rumex patientia L. x Rumex tianschanicus A.Los) and applied to H2O2-treated HepG2 cells to investigate their antioxidant properties.
At low concentrations, both peptides were not cytotoxic to HepG2 cells, reduced the production of intracellular ROS and MDA, and increased the enzymatic activities of SOD, GSH-Px, and CAT.
The potential applications of food-derived antioxidant peptides as additives, nutraceuticals and therapeutic agents have fuelled the current interest in discovering them from different plant sources.
The authors also mention the potential use of two peptides studied as food additives or supplements. Whether this is the purpose of finding antioxidant peptides in the RRL needs to be better explained and discussed.
The topic is interesting and timely, but the paper has some serious gaps.
The work is sloppily written. Some examples.
- Remove from the title the words: synthesis of 2 plant peptides. The authors, in fact, identify the peptides and determine their antioxidant activity but do not report the synthesis of the peptides in the methods.
The author's answer: We sincerely thank the reviewer for careful reading. As suggested by the reviewer. We have removed the word "synthesis" from the title.
- The name of the so called “protein grass” is spelled wrong in the title, in abstract, and in lines 27, 68 etc. (Rumexpatientia L. x Rumextianshanicus A. Losinstead of Rumex patientia L. x Rumex tianschanicus A.Los).
The author's answer: It is really a giant mistake to the whole quality of our article. We feel sorry for our carelessness. We have corrected the description in the full text and we also feel great thanks for your point out.
- Authors must specify what kind of neutral protease they use in materials and methods.
The author's answer: We sincerely appreciate the valuable comments. We have added the relevant content in " Materials and Methods " and changed the text to red to make it easier for you to find. Neutral protease is obtained by fermentation of Bacillus subtilis.
- The authors say that in China the leaves, flowers, and seeds of certain Rumex plant species are consumed as food. For the hydrolysis of RRL proteins, however, the authors use a neutral protease and not a standard gastrointestinal digestion method. They should explain why they use a neutral protease (perhaps to prepare peptides for industrial purpose?).
The author's answer: We appreciate your question. Neutral protease was chosen because of its mildness and high extraction rate in the previous study, while pepsin and trypsin hydrolyzed the peptide at a lower extraction rate. Neutral proteases were also chosen with a view to the subsequent production of industrial peptides.
- Furthermore, the authors say that RRL, known as "protein grass “, in China, was recognized as a “new food ingredient” in 2021. So, is it an ingredient or a food?
The author's answer: Thank you again for your positive comments and valuable suggestions to improve the quality of our manuscript. In China, the term "new food ingredients" refers to the following items that are not traditionally consumed in China: animals, plants and micro-organisms; ingredients isolated from animals, plants and micro-organisms; food ingredients that have undergone changes in their original structure; and other newly developed food ingredients. RRL is a food product that has been implemented according to the Chinese standard for vegetables and is also used as an ingredient in other food products.
- Considering these observations, the entire discussion must be rewritten, better explaining the objectives that the authors set themselves and the prospects. What is known about the efficacy of these peptides or RRL in vivo? The impact of gastrointestinal digestion on the efficacy, stability, and biological availability of these peptides?
The author's answer: We have tried our best to improve the manuscript and have made some changes to the manuscript. These changes will not influence the content and framework of the paper. And here we did not list the changes but marked in red in the revised paper. We appreciate for Editors/Reviewers’ warm work earnestly and hope that the correction will meet with approval.
- Figures 6 and 7 are difficult to read and difficult to understand. They need to be redone and their legends rewritten. For example, Fig. 6 legend: Figure 6. Effect of peptides P1 and P3 on cell viability in the presence of H2O2. a) Effect of different concentrations of H2O2 on cell survival. b) Effect of different concentrations of P1 and P3 on cell viability. c) Protective effect of different concentrations of P1 and P3 from H2O2 damage. d) Effect of different concentrations of P1 and P3 on ROS level.
The author's answer: We think this is an excellent suggestion. We have rewritten this part as suggested by the reviewer. We have optimized the images in Figures 6 and 7, while the legends have been improved.
We tried our best to improve the manuscript and made some changes marked in red in revised paper which will not influence the content and framework of the paper. We appreciate for Editors/Reviewers’ warm work earnestly and hope the correction will meet with approval. Once again, thank you very much for your comments and suggestions.
Yours sincerely,
Chang Liu
11 March 2024
School of Food and Health
Beijing Technology and Business University
Beijing 100048, China
E-mail: lc15373689836@163.com
aijinma206@sina.com
Reviewer 3 Report
Comments and Suggestions for Authors
The article titled “Preparation, isolation, synthesis and antioxidant function of 2 plant peptides from a new resource of Rumexpatientia L. ×Ru-3 mextianshanicus A. Los” presents novel results, since it studies the potential peptides responsible for antioxidant activity encrypted within RRL proteins. On the other hand, the methodology used and the discussion carried out is appropriate.
I recommend approval of the manuscript after minor revisions.
Introduction
Line 48-50 RLL contains all essential amino acid, but Does it not contain limiting amino acid according to reference protein for adults (FAO 2007)?
Materials and Methods
Line 208-211. Check and correct the narrative (authors should use passive voice in past)
Comments on the Quality of English Language
No comments
Author Response
Dear Editors and Reviewers:
Thank you for your letter and for the reviewers’ comments concerning our manuscript entitled “Preparation, isolation and antioxidant function of peptides from a new resource of Rumexpatientia L. x Rumextianshanicus A. Los” (ID: foods-2901464). The comments are all valuable and very helpful for revising and improving our paper, as well as the important guiding significance to our research. We have studied comments carefully and have made correction which we hope meet with approval.
Response to Reviewer 3 Comments
Reviewer #3 The article titled “Preparation, isolation, synthesis and antioxidant function of 2 plant peptides from a new resource of Rumexpatientia L. ×Rumextia-nshanicus A. Los” presents novel results, since it studies the potential peptides responsible for antioxidant activity encrypted within RRL proteins. On the other hand, the methodology used and the discussion carried out is appropriate. I recommend approval of the manuscript after minor revisions.
- Introduction: Line 48-50 RLL contains all essential amino acid, but Does it not contain limiting amino acid according to reference protein for adults (FAO 2007)?
The author's answer: We think this is an excellent suggestion. We have added the relevant content of restricted amino acids in lines 47-55 of the revised draft, and at the same time integrated it with the relevant content of the FAO/WHO regulations, and marked it in red.
- Materials and Methods: Line 208-211. Check and correct the narrative (authors should use passive voice in past)
The author's answer: We have tried our best to improve the manuscript and made some changes to the manuscript. These changes will not influence the content and framework of the paper. And here we did not list the changes but marked in red in the revised paper. We appreciate for Editors/Reviewers’ warm work earnestly and hope that the correction will meet with approval.
We tried our best to improve the manuscript and made some changes marked in red in revised paper which will not influence the content and framework of the paper. We appreciate for Editors/Reviewers’ warm work earnestly and hope the correction will meet with approval. Once again, thank you very much for your comments and suggestions.
Yours sincerely,
Chang Liu
11 March 2024
School of Food and Health
Beijing Technology and Business University
Beijing 100048, China
E-mail: lc15373689836@163.com
aijinma206@sina.com

Round 2
Reviewer 1 Report
Comments and Suggestions for Authors
The authors have made significant changes to prepare the document "Preparation, isolation and antioxidant function of peptides 2 from a new resource of Rumexpatientia L. x Rumextianshanicus A. Los." However, the manuscript has some details that need improvement.
Below are my comments:
Line 84 and subsequent. Insert a space between the magnitude and the units (g, °C, L).
Line 146. Change "minutes word" to "min."
Line 204 and subsequent. Use "h" instead of "hours."
Line 275. The IC50 reported by Ahmidin et al. should appear as 7.8 ± 0.09 mg/mL instead of 7.8 ± 0.09.
Lines 264-265. Review the IC50 values for the scavenging activities method because they do not match with results showed in Fig 1.
Figure 4. Include in the figure caption the meaning of each label (S1, S2, S3, RH).
Lines 322-332. Why wasn't the hydroxyl radicals method performed for S3G1 and S3G2? Please explain.
Figure 3. Use the same graph style (as in figures 1 and 2) to report the results of the DPPH method.
Figure 5. The resolution of Figure 5 is very poor; please improve the image. In figures c and d, indicate which is P1 and which is P3. In the figure caption, include the Keap1 access code (PDB: 2FLU).
Lines 374 and 375. Review the wording.
Tables 3 and 4. In the name of amino acids (three-letter code), use uppercase and lowercase. For example, Tyr instead TYR.
In the docking, please include the docking energy to identify the stability of the interaction between Keap1-P1 and Keap1-P3.
Lines 474-489. It is necessary to compare the results obtained in this study with some other similar studies.
I suggest that section 3 (Results) be titled "Results and Discussion" and that the current discussion section be titled "Conclusions."
Author Response
Dear Editors and Reviewers:
Thank you for your letter and the reviewers’ comments on our manuscript entitled “Preparation, isolation and antioxidant function of peptides from a new resource of Rumexpatientia L. x Rumextianshanicus A. Los” (ID: foods-2901464). Those comments are very helpful for revising and improving our paper, as well as the important guiding significance to other research. We have studied the comments carefully and made corrections which we hope meet with approval. The main corrections are in the manuscript and the responds to the reviewers’ comments are as follows.
Response to Reviewer 1 Comments
Reviewer #1: The authors have made significant changes to prepare the document "Preparation, isolation and antioxidant function of peptides 2 from a new resource of Rumexpatientia L. x Rumextianshanicus A. Los." However, the manuscript has some details that need improvement.
- Line 84 and subsequent. Insert a space between the magnitude and the units (g, °C, L).
The author's answer: Many thanks indeed for pointing this out, we have corrected it in the full text.
- Line 146. Change "minutes word" to "min."
The author's answer: We sincerely thank the reviewer for careful reading. As suggested by the reviewer, we have corrected the “minutes” into “min”.
- Line 204 and subsequent. Use "h" instead of "hours."
The author's answer: We were really sorry for our careless mistakes. Thank you for your reminder.
- Line 275. The IC50 reported by Ahmidin et al. should appear as 7.8 ± 0.09 mg/mL instead of 7.8 ± 0.09.
The author's answer: We have re-written this part according to the Reviewer’s suggestion, replace it with "7.8 ± 0.09 mg/mL" in line 274 and highlighted it in yellow.
- Lines 264-265. Review the IC50 values for the scavenging activities method because they do not match with results showed in Fig 1.
The author's answer: Thank you again for your positive comments and valuable suggestions to improve the quality of our manuscript. We reconfirmed the pre-IC50 data and placed them in Table 1, which was calculated by the software GraphPad Prism 8, which indeed showed IC50 values for ABTS radicals, DPPH radicals, and hydroxyl radicals of 174.7 ± 1.91 μg/mL, 260.3 ± 4.88 μg/mL, and 1.86 ± 0.07 mg/mL, respectively.
Table1 Raw data on IC50 of ABTS radicals, DPPH radicals and hydroxyl radicals by RRL hydrolyzed peptides
|
Free radicals |
Concentration (μg/mL) |
Data 1 (%) |
Data 2 (%) |
Data 3 (%) |
Average value (%) |
Error (%) |
IC50 (μg/mL) |
|
ABTS |
100 |
33.93274 |
29.99369 |
34.46121 |
32.79588 |
2.441114 |
174.7 |
|
200 |
49.26726 |
55.19315 |
51.50482 |
51.98841 |
2.9924 |
||
|
300 |
67.16062 |
69.65607 |
65.69631 |
67.50433 |
2.002133 |
||
|
400 |
79.23333 |
76.82857 |
79.55714 |
78.53968 |
1.490684 |
||
|
500 |
84.94864 |
85.05457 |
84.45746 |
84.82022 |
0.318594 |
||
|
DPPH |
100 |
25.72535 |
24.44366 |
24.86268 |
25.01056 |
0.653518 |
260.3 |
|
200 |
45.65141 |
42.98944 |
41.93662 |
43.52582 |
1.914601 |
||
|
300 |
49.08099 |
53.02817 |
48.57746 |
50.22887 |
2.437299 |
||
|
400 |
60.52700 |
63.60798 |
62.67958 |
62.27152 |
1.580507 |
||
|
500 |
72.88732 |
68.97183 |
71.82629 |
71.22848 |
2.025044 |
||
|
OH |
1000 |
26.5385 |
27.8359 |
27.71344 |
27.36261 |
0.716324 |
1856 |
|
2000 |
53.5821 |
54.4428 |
47.7465 |
51.9238 |
3.643157 |
||
|
3000 |
69.7421 |
64.6848 |
68.14308 |
67.52333 |
2.584984 |
||
|
4000 |
79.3851 |
82.4786 |
72.6363 |
78.16667 |
5.033006 |
||
|
5000 |
87.3857 |
88.2723 |
91.8635 |
89.17383 |
2.371127 |
- Figure 4. Include in the figure caption the meaning of each label (S1, S2, S3, RH).
The author's answer: We think this is an excellent suggestion. We have rewritten this section as suggested by the reviewer. The caption of Figure 4, which we have included, explains what each label mean and highlighted it in yellow, and we appreciate you pointing this out.
- Lines 322-332. Why wasn't the hydroxyl radicals method performed for S3G1 and S3G2? Please explain.
The author's answer: Thank you for your question, the fractions of S3G1 and S3G2 are shown by the graph of gel filtration in Fig. 3(a), the S3G2 fraction was obtained less, the ratio of the two fractions was 99.05:0.95, and only 18 mg of the S3G2 fraction was obtained in the two months of extraction during the research process, and it may not be possible to measure the hydroxyl radicals.
- Figure 3. Use the same graph style (as in figures 1 and 2) to report the results of the DPPH method.
The author's answer: We sincerely appreciate your valuable comments. We have changed the graph on DPPH radicals in Figure3(c) from a bar graph to a line graph, consistent with the legends in Figures 1 and 2.
- Figure 5. The resolution of Figure 5 is very poor; please improve the image. In figures c and d, indicate which is P1 and which is P3. In the figure caption, include the Keap1 access code (PDB: 2FLU).
The author's answer: Thanks to the reviewers for pointing this out. We have made changes. We have improved the resolution of the image in Figure 5, added the annotations "P1" and "P3" in Figure 5(c) and (d), and eventually added the PDB of Keap1 (PDB: 2FLU) to the title of Figure 5.
- Lines 374 and 375. Review the wording.
The author's answer: We apologize for any negligence in our written narrative. The relevant content has now been amended and highlighted in yellow on lines 372-374.
- Tables 3 and 4. In the name of amino acids (three-letter code), use uppercase and lowercase. For example, Tyr instead TYR.
The author's answer: It is really a giant mistake to the whole quality of our article. We feel sorry for our carelessness. We have corrected the description in the full text and we also feel great thanks for your point out.
- In the docking, please include the docking energy to identify the stability of the interaction between Keap1-P1 and Keap1-P3.
The author's answer: Considering the Reviewer's suggestion, we have filled in the docking energies of P1 and P3 to lines 383 and 386, which were obtained using the AutoDock VINA software, as indicated in lines 190 and 191 of the method.
- Lines 474-489. It is necessary to compare the results obtained in this study with some other similar studies.
The author's answer: Thank you for your valuable suggestions to improve the readability of our manuscript. We have added other similar studies in lines 490-494 for your reference.
- I suggest that section 3 (Results) be titled "Results and Discussion" and that the current discussion section be titled "Conclusions."
The author's answer: We appreciate your detailed review. We have changed the titles of both parts of the manuscript.
We tried our best to improve the manuscript and made some changes to the manuscript. These changes will not influence the content and framework of the paper. And here we did not list the changes but marked in yellow in the revised paper. We appreciate for Editors/Reviewers’ warm work earnestly and hope that the correction will meet with approval.
Yours sincerely,
Chang Liu
17 March 2024
School of Food and Health
Beijing Technology and Business University
Beijing 100048, China
E-mail: lc15373689836@163.com
aijinma206@sina.com
Reviewer 2 Report
Comments and Suggestions for Authors
The authors made all the required corrections, and the paper is significantly improved. In order to be published on Foods the discussion must be improved and broadened as it appears a summary of the work carried out and with few observations. I suggest that the authors delve deeper into the discussion on some aspects such as, for example, the possible use of these peptides which is just mentioned (see lines 506-509) and also the comparison with other peptides etc... An explanation of the advantages of choosing a neutral protease for producing peptides industrially could also be added to the discussion.
Author Response
Dear Editors and Reviewers:
Thank you for your letter and the reviewers’ comments on our manuscript entitled “Preparation, isolation and antioxidant function of peptides from a new resource of Rumexpatientia L. x Rumextianshanicus A. Los” (ID: foods-2901464). Those comments are very helpful for revising and improving our paper, as well as the important guiding significance to other research. We have studied the comments carefully and made corrections which we hope meet with approval. The main corrections are in the manuscript and the responds to the reviewers’ comments are as follows.
Response to Reviewer 2 Comments
Reviewer #2: The authors made all the required corrections, and the paper is significantly improved. In order to be published on Foods the discussion must be improved and broadened as it appears a summary of the work carried out and with few observations. I suggest that the authors delve deeper into the discussion on some aspects such as, for example, the possible use of these peptides which is just mentioned (see lines 506-509) and also the comparison with other peptides etc... An explanation of the advantages of choosing a neutral protease for producing peptides industrially could also be added to the discussion.
The author's answer: We sincerely thank the editor and all reviewers for their valuable comments, which we utilized to improve the quality of the manuscript. Our responses are listed in normal font and additions to the manuscript are highlighted in yellow. We add the advantages of selecting neutral proteases for industrial production of peptides at 512-518, and the possible uses of RLL peptides and possible reasons for functional food additives at 527-538. The importance of gastrointestinal digestion and subsequent cellular pathways, animal models, and clinical studies is also added in 539-562 with examples.
We tried our best to improve the manuscript and made some changes to the manuscript. These changes will not influence the content and framework of the paper. And here we did not list the changes but marked in yellow in the revised paper. We appreciate for Editors/Reviewers’ warm work earnestly and hope that the correction will meet with approval.
Yours sincerely,
Chang Liu
17 March 2024
School of Food and Health
Beijing Technology and Business University
Beijing 100048, China
E-mail: lc15373689836@163.com
aijinma206@sina.com